# Retrospective and Comparative Study of Three Molecular Assays for the Macrolide Resistance Detection in *Mycoplasma genitalium* Positive Urogenital Specimens

**DOI:** 10.3390/ijms24087218

**Published:** 2023-04-13

**Authors:** María Paz Peris, Blanca Dehesa, Henar Alonso, Cristina Escolar, Laura Clusa, Miriam Latorre-Millán, Antonio Rezusta, Ana Milagro

**Affiliations:** 1Instituto de Investigación Sanitaria Aragón, 50009 Zaragoza, Spain; lauraclusa@gmail.com (L.C.); mlatorre@iisaragon.es (M.L.-M.); arezusta@salud.aragon.es (A.R.); amilagro@salud.aragon.es (A.M.); 2Department of Animal Pathology, Faculty of Veterinary, University of Zaragoza, 50013 Zaragoza, Spain; 3Department of Microbiology, Pediatrics, Radiology, and Public Health, Faculty of Medicine, University of Zaragoza, 50009 Zaragoza, Spain; dgblanca7@gmail.com (B.D.); henar83.alonso@gmail.com (H.A.); 4Department of Animal Production and Food Science, Faculty of Veterinary, University of Zaragoza, 50013 Zaragoza, Spain; cristinaesco87@gmail.com; 5Miguel Servet University Hospital, Microbiology, 50009 Zaragoza, Spain

**Keywords:** *Mycoplasma genitalium*, macrolide resistance, real-time PCR, rapid diagnostic

## Abstract

The capacity of *Mycoplasma genitalium* to develop resistance to macrolides makes detection of macrolide resistance genes by rapid real-time PCR assays increasingly necessary in clinical diagnostic laboratories so as to initiate appropriate treatment as rapidly as possible. The aim of this retrospective and comparative study was to conduct the clinical evaluation of three commercially available kits for macrolide resistance detection. A total of 111 *M. genitalium* positive samples analyzed in the Clinical Microbiology Laboratory of the Miguel Servet University Hospital, Zaragoza (Spain) were used. After *M. genitalium* molecular confirmation, the three assays under study were evaluated and discrepant results were resolved via sequencing. The clinical sensitivity for resistance detection was 83% (95% confidence interval, 69% to 93%) for the ResistancePlus^®^ MG panel kit (SpeeDx Pty Ltd., Sydney, Australia), 95% (84% to 99%) for Allplex^TM^ MG & AziR Assay (Seegene^®^, Seoul, Korea), and 97% (88% to 99%) for the VIASURE macrolide resistance-associated mutations (23SrRNA) Real time PCR detection kit (Certest Biotec, Zaragoza, Spain). The clinical specificity was 100% (94% to 100%) for Allplex and VIASURE assays and 95% (86% to 99%) for SpeeDx assay. The results arising from this study are cause for strong consideration for the implementation of rapid real-time PCR assays in clinical diagnosis laboratories to eliminate treatment failure and transmission as soon as possible.

## 1. Introduction

*Mycoplasma genitalium* is a sexually transmitted infection (STI) recognized as cause of urethritis and further urogenital syndromes. The clinical importance of *M. genitalium* infection has been recognized by the Centers for Disease Control and Prevention, who list *M. genitalium* under “Emerging Issues” in the 2015 Sexually Transmitted Disease Treatment Guidelines [1].

This bacterium has a small genome and lacks a cell wall, which reduces the effectiveness of conventional antibiotics, in particular penicillins and other beta-lactam antibiotics. Treatment options are limited to antimicrobials that interrupt protein synthesis (e.g., tetracyclines, macrolides, streptogramins) or DNA replication (e.g., fluoroquinolones) [2].

*Mycoplasma genitalium* has a distressing capacity to develop resistance to macrolides and fluoroquinolones, with macrolide resistance being the most prevalent [1,3]. Macrolide resistance is rapidly increasing worldwide, ranging from 44% to 90% in the United States, Canada, Western Europe, and Australia [1,2,3,4]. Failure of azithromycin treatment is strongly associated with the presence of point mutations in the V-binding region of the 23S rRNA gene in 2058 and 2059 positions (Escherichia coli numbering) of *M. genitalium* [5,6]. The prevalence of quinolone resistance markers is significantly lessened and treatment failure with moxifloxacin is predominantly mediated by key mutations in the quinolone resistance revealing region of the topoisomerase IV parC gene, usually at amino acid positions S83 and D87 (*M. genitalium* numbering) [7,8].

Regarding the diagnostic technique, DNA sequencing is the gold standard method for mutant determination [8,9]. Recent studies have used sequencing for the detection of a fragment of the V-binding region of the 23S rRNA gene of *M. genitalium* [10,11,12,13,14] or rapid pyrosequencing assay [15]. Sequencing, however, is generally not feasible for routine diagnosis due to the higher cost and longer turnaround time to results, postponing the reporting of macrolide resistance and, hence, treatment with second-line antimicrobials [16]. This leads to rapid real-time PCR assays in clinical laboratories, providing high sensitivity, specificity, and speediness in obtaining the result, with favorable cost-effectiveness [17].

In view of this situation, the objective of this work was to determine the diagnostic accuracy of three available rapid real-time PCR assays for macrolide resistance gene detection in positive *M. genitalium* clinical samples: ResistancePlus^®^ MG panel kit (SpeeDx Pty Ltd., Sydney, Australia) (SpeeDx assay), Allplex^TM^ MG & AziR Assay (Seegene^®^, Seoul, Korea) (Allplex assay), and the VIASURE macrolide resistance-associated mutations (23SrRNA) Real time PCR detection kit (Certest Biotec, Zaragoza, Spain) (Viasure assay).

## 2. Results

### 2.1. Samples and Tests Results

A total of 111 clinical samples were analyzed: 54 urethral swabs, 9 rectal swabs, 39 endocervical swabs, 3 vaginal swabs, and 6 urethral urines.

Viasure and Allplex assays yielded valid results for all samples studied. However, when analyzing SpeeDx assay results, five samples were reported as inconclusive, even after repetition. In these cases, instructions for use (IFUs) recommend new specimen sampling. Thus, these five samples were excluded from SpeeDx analyses. The macrolide resistance percentage from the samples studied varied depending on the molecular assay used: SpeeDx yielded 36.7%, Allplex 41.4%, and Viasure 42.3% samples with macrolide point mutations (Table 1).

No mutations were detected by any of the assays under study in 56 *M. genitalium* positive samples. These samples were considered sensitive to macrolides. Similarly, 33 *M. genitalium* positive samples presented point mutations and were therefore considered resistant to macrolides. A total of 17 samples yielded discrepant results among the three assays. Briefly, three samples were reported as resistant for SpeeDx but sensitive for Viasure and Allplex assays. In a similar manner, 11 sensitive samples for SpeeDx were considered macrolide resistant by Viasure and Allplex assays. Viasure assay yielded one sample as sensitive that tested resistant by SpeeDx and Allplex assays. Allplex assay yielded two sensitive samples considered resistant by SpeeDx and Viasure assays. In regard to the five inconclusive results for SpeeDx, when tested by Viasure and Allplex assays they were reported as one resistant and four sensitive.

Neither SpeeDx nor Viasure assays specify the mutation, however, Allplex assay provides the specific single-nucleotide polymorphism (SNP) responsible for resistance. The Allplex assay results for specific point mutations were: sixteen samples for A2058C, nine samples for A2058G, seven samples for A2059C, fourteen samples for A2059G, and zero samples for A2059T and A2058C mutations.

### 2.2. Discrepant Sample Resolution

Discrepant and inconclusive SpeeDx samples were consequently sequenced. Out of these inconclusive samples, four could not be sequenced since no band was obtained on agarose gel and were hence removed from the study. Discrepant sample results are shown in Table 2.

Finally, after discrepant sample resolution, a total of 44/107 samples were considered positive for macrolide resistance mutations and 63/107 were considered sensitive. The prevalence obtained was 41.1% (CI = 31.8–51).

### 2.3. Clinical Validation

For clinical validation calculations, inconclusive SpeeDx assay samples were not considered because no new samples could be obtained. Finally, SpeeDx assay yielded three false positive and seven false negative results, Allplex assay two false negative results, and Viasure assay one false negative result for point mutation detection (Table 3).

## 3. Discussion

This comparative–retrospective observational study yielded valid and relevant clinical accuracy results for the three assays under study.

Several studies have been previously performed to evaluate the SpeeDx assay clinical accuracy. These studies have yielded both similar and different results to the ones obtained in this evaluation. For example, higher sensitivity values have been reported: 95.4% [18], 97.4% [19], and 100% [16,20]. On the other hand, Naidu et al. (2021) [21] observed low sensitivity values (71.4%). The moderate sensitivity of the SpeeDx assay observed in this evaluation (83%) is concerning, since 6.6% (7/106) of the specimens yielded a false negative result for macrolide resistance. This suggests a risk of treatment failure if azithromycin were subsequently administered. The specificity for macrolide resistance detection of 95% found is in accordance with those reported in previous studies: 94.6% [20], 95.8% [18], 96.2% [16], and 100% [19,21].

Regarding Allplex assay, previous studies yielded lower sensitivity values (74.5% and 71.4%) [21,22] than obtained in this evaluation (95%), however, specificity values are in consonance with our results: 97.6% [22] and 100% [21].

To the authors’ knowledge, there are no scientific publications validating VIASURE assay with clinical samples. Highly accurate sensitivity (97%) and specificity (100%) values were obtained.

In terms of mutations, all the assays under study detect point mutations at positions 2058 and 2059 associated with failure of azithromycin treatment [5,6]. Both SpeeDx and Viasure assays detect five macrolide resistance mutations, reporting the results as either mutant presence or absence. However, Allplex assay detects six macrolide resistance mutations and provides the detected single-nucleotide polymorphism in the result. Treatment is non-reliant on the mutation detected, but this can be useful for surveillance or research purposes. Fortunately, Seegene assay detected no A2059T mutation (mutation not detected by SpeeDx and Viasure assays); however, in the case of this type of mutation, SpeeDx and VIASURE kits would yield false negative results for macrolide resistance.

On the other hand, although with low frequency, mutations at position 2062 have been previously observed (2.3% and 7.7% for A2062T and A2062C mutations) [13,23]. Mutations at position 2062 have been associated with macrolide and streptogramine treatment failure, such as josamycin (A2062G) and pristinamycin (A2062T) [24,25]. None of the assays evaluated in this study are designed to detect mutations at this position, which might have led us to misconstrue specimens harboring mutated *M. genitalium* as previously reported [22]. Le Roy et al. (2021) evaluated the clinical performance of three kits: Allplex^TM^ MG & AziR Assay (Seegene^®^, Seoul, Korea), the Macrolide-R/MG ELITe MGB (ELITechGroup, Puteaux, Hercules, CA USA), and the ResistancePlus MG FleXible (SpeeDx Pty Ltd, Sydney, Australia) and 7.3% of the samples showed A2062T mutation, only detected by sequencing. This suggests that up to 7.3% of *M. genitalium* positive specimens could be falsely considered as sensitive/wild type. In this study, only positive discordant samples, previously analyzed by the three kits, were sequenced. In the case of 2062 mutation, it was not detected by any of the kits under study and, consequently, as they were not sequenced, they could not have been falsely categorized as sensitive/wild type.

As for the time required to assemble a PCR plate, it is much shorter for the Viasure assay than for the Seegene and SpeeDx assays. VIASURE reagents are freeze-dried in a stabilized format (containing a mix of enzymes, primers probes, buffer, dNTPs, stabilizers, and internal control) allowing greater stability in the 2 °C to 40 °C temperature range. This leads to less manipulation and, therefore, less handling error. By contrast, the reagents for the Allplex and the SpeeDx assays are in a separate liquid state requiring more hands-on time.

To perform amplification, Allplex assay strictly requires the CFX96 Real Time PCR System. However, SpeeDx can be used on any instrument supporting Taqman technology and Viasure assay in any instrument supporting real-time PCR technology, making them less reliant on equipment. While SpeeDx and Allplex assays require specific interpretation software to establish their results correctly, Viasure assay allows for interpretation using the instrument itself. Thus, it is our recommendation that the selection of the kit should be made according to the laboratory resources.

The resistance prevalence obtained during this study was higher than those previously reported in Spain. This study reports a value of 41.1%, while other past studies have reported: 36.1% in Barcelona [26], 36.4% in Granada [27], and 20% in Madrid [11]. However, no reliable prevalence data can be extracted from our results because clinical information could not be accessed, and we were unable to determine if the samples were from symptomatic persons and/or if they had received previous treatment. Similarly, the high prevalence found in this study and the lack of studies in Aragón (Spain) encourage future studies focused on the detection of macrolide resistance genes. Furthermore, more accurate results could be obtained if the study was prospective and multicentric rather than retrospective and single-centered.

## 4. Materials and Methods

### 4.1. Study Design

This was a comparative and retrospective study using *M. genitalium* positive DNA samples previously diagnosed using a commercially available real-time PCR kit. The three kits under study were then tested for resistance or sensitivity to macrolide antibiotics.

The algorithm used for the clinical analysis was: (i) point mutation positive results from the three assays were considered true positive values (macrolide-resistant samples); (ii) point mutation negative results from the three assays were considered as true negative values (sensitive or wild type samples); (iii) discrepant results were resolved by sequencing.

### 4.2. Participants and Samples

The study was conducted with a total of 111 *M. genitalium* positive urogenital samples collected from January 2021 to March 2022, from patients tended to in centers and hospitals that were referred to the Clinical Microbiology Laboratory from the Miguel Servet University Hospital, Zaragoza (Spain). The bacterium was previously detected during routine laboratory diagnosis using the available commercial assay Allplex™ STI essential assay panel (Seegene^®^, Seoul, Korea), by real-time PCR in the CFX96 ™ real-time PCR system (Bio-Rad^®^ Laboratories, Hercules, CA, USA) and the results were analyzed using the Seegene software (Seegene Viewer for Realtime instruments V.3 3.17.000). This assay can simultaneously detect seven pathogens in a single well: *Ureaplasma urealyticum*, *Neisseria gonorrhoeae*, *Mycoplasma hominis*, *Mycoplasma genitalium, Ureaplasma parvum*, *Chlamidia trachomatis*, and *Trichomonas vaginalis*. For this study, only those positive for *M. genitalium* were considered.

DNA extractions were performed using 200 μL of the clinical samples with the Mag Lead^®^ 12gC automatic DNA extraction system (PSS biosystem, Pleasanton, CA, USA) according to the manufacturer’s instructions. All nucleic acid elutions were performed in 50 μL of elution buffer. Once the DNAs were extracted and used in the clinical diagnosis, the remnant was stored appropriately at −20 °C until used for molecular analysis.

### 4.3. Test Methods

#### 4.3.1. ResistancePlus^®^ MG Panel Kit (SpeeDx, Sydney, Australia): SpeeDx Assay

This assay detects *M. genitalium* and five 23S rRNA gene mutations: A2058G, A2059G, A2059C, A2058T, and A2058C (Table 4), and can be performed on any instrument supporting Taqman technology. This assay was analyzed with a Cobas Z 480 PCR Analyzer IVD LightCycler 480 II from Roche and the batch used in this study was 21020005 (expiry date: 30 April 2022). Following the instructions for use (IFUs), color compensation was performed by using the PlexPCR Colour Compensation (SpeeDx Pty Ltd., Sydney, Australia) (batch: 21010015, expiry date: 1 October 2022). Once all runs were obtained the analyses were interpreted with the UgenTec FastFinder 3.5.8 from SpeeDx software. This assay yields the *M. genitalium* presence or absence and the presence of point mutations. When the test does not detect any of the point mutations, the sample is reported as *M. genitalium* susceptible. Samples with inconclusive values were repeated in accordance with IFU results interpretation stipulations.

#### 4.3.2. Allplex^TM^ MG & AziR Assay (Seegene^®^, Seoul, Korea): Allplex Assay

Similarly to the former, this assay allows simultaneous amplification and detection of *M. genitalium* target nucleic acids and differentiation of six 23S rRNA gene mutations: A2058G, A2058C, A2058T, A2059G, A2059C, and A2059T (Table 4). The batch used in this study was SDC221K06 (expiry date: 31 October 2022). This analysis was performed in the CFX96 ™ real-time PCR system (Bio-Rad^®^ Laboratories, Hercules, CA, USA) required by the IFUs, and the results were analyzed using the Seegene software (Seegene Viewer for Realtime instruments V.3 3.17.000). This assay provides the *M. genitalium* presence and the specific SNP responsible for resistance.

#### 4.3.3. VIASURE Macrolide Resistance-Associated Mutations (23SrRNA) Real Time PCR Detection Kit (Certest Biotec, Zaragoza, Spain): Viasure Assay

This assay is designed for the detection and differentiation of the presence of the V-binding wild type (WT) region of the 23S rRNA gene or the presence of point mutations of the same region associated with macrolide resistance (point mutations detected: A2058G, A2059G, A2058T, A2058C, and A2059C) in *M. genitalium* positive samples (Table 4). It should be noted that this region is common for the pathogens *M. genitalium* and *M. pneumoniae*, the Mycoplasma species phylogenetically closest to *M. genitalium* [28].

The batch used in this study was MGR1XH-Exp.620B (expiry date: 04/24). This assay can be performed on any instrument supporting real-time PCR technology. In particular, this analysis was performed in the CFX96 ™ real-time PCR system (Bio-Rad^®^ Laboratories, France), and the results were interpreted with Bio-Rad CFX ™ Manager IVD Edition 1.6.

This assay yields the presence of the WT region (macrolide-sensitive samples) and the presence of point mutation (macrolide-resistant samples). Following IFU results interpretation, samples with invalid results were 1:10 diluted and repeated.

#### 4.3.4. Sequencing

Samples with discrepant results were sequenced. In brief, a 321 pb conserved region of the 23S rRNA gene was amplified using the forward primer 5′- TGTATATGGGGTGACACCTG -3′, reverse primer 5′- AATCCTTGCGAACTTGCATC -3′. For purified PCR amplicons and bidirectional Sanger sequencing, the non-purified PCR product together with the PCR forward and reverse primers (both at 25 pmol) were sent to STAB VIDA, Lda (Caparica, Portugal). Obtained sequences were edited manually, by means of the CodonCode Aligner software (CodonCode Corporation, Dedham, MA, USA). A BLAST examination (https://blast.ncbi.nlm.nih.gov/Blast.cgi, accessed on 4 July 2022) was performed using the obtained nucleotide sequences for species classification.

### 4.4. Analyses

The data were collected in an Excel file including all results. Clinical sensitivity, specificity, and negative and positive predictive values (with 95% confidence intervals) were calculated using the MetaDisc v1.4 freeware software [29].

The minimum sample size was calculated with WinEpi 2.0 (http://www.winepi.net/winepi2/, accessed on 23 May 2022) [30] with the estimate proportion (random sampling & perfect diagnostic) option. From an unknown population size, and using calculation based on a normal distribution, the minimum sample size calculated was 89 individuals to analyze an estimated proportion of 36% with an accepted error (or precision) of 10% and a confidence level of 95%.

## 5. Conclusions

The findings obtained in this study lead to considering the implementation of rapid real-time PCR assays in clinical diagnostic laboratories allowing high sensitivity and specificity values. The high prevalence obtained in the studied samples encourages this implementation to avoid treatment failure and stop transmission as soon as possible.

## Figures and Tables

**Table 1 ijms-24-07218-t001:** ResistancePlus^®^ MG panel kit (SpeeDx assay), Allplex^TM^ MG & AziR Assay (Allplex assay), and VIASURE macrolide resistance-associated mutations (23SrRNA) Real time PCR detection kit (Viasure assay) results.

	SpeeDx Assay N Detected/N Total (%)	Allplex Assay N Detected/N Total (%)	Viasure Assay N Detected/N Total (%)
Point mutations	39/106 * (36.7%)	46/111 (41.4%)	47/111 (42.3%)
Sensitive/WT	67/106 * (63.2%)	65/111 (58.5%)	64/111 (57.6%)
Inconclusive	5/111 (4.5%)	0/111 (0%)	0/111 (0%)

* N total—5 inconclusives (as previously described); WT: Wild type.

**Table 2 ijms-24-07218-t002:** Discrepancies among ResistancePlus^®^ MG panel kit (SpeeDx), VIASURE macrolide resistance-associated mutations (23SrRNA) Real time PCR detection kit (Viasure), and AllplexTM MG & AziR Assay (Allplex) and sequencing results.

ID	Sample Type	SpeeDx	Allplex	Viasure	Sequencing
1	Urethral	Point mutations	Sensitive	Sensitive	Sensitive
2	Endocervical	Inconclusive	Sensitive	Sensitive	Sensitive
3	Endocervical	Inconclusive	Sensitive	Sensitive	Sensitive
4	Urethral	Sensitive	Point mutation	Point mutations	N/D *
5	Rectal	Sensitive	Point mutations	Point mutations	Point mutation
6	Rectal	Sensitive	Point mutation	Point mutations	Point mutation
7	Urethral	Sensitive	Point mutations	Point mutations	Point mutation
8	Urethral	Point mutations	Point mutation	Sensitive	Point mutation
9	Urethral	Sensitive	Point mutation	Point mutations	N/D *
10	Urethral	Point mutations	Sensitive	Sensitive	Sensitive
11	Endocervical	Sensitive	Point mutation	Point mutations	Point mutation
12	Endocervical	Sensitive	Point mutation	Point mutations	Point mutation
13	Urethral	Sensitive	Point mutation	Point mutations	Point mutation
14	Endocervical	Inconclusive	Sensitive	Sensitive	Sensitive
15	Urethral	Point mutations	Sensitive	Sensitive	Sensitive
16	Urethral	Sensitive	Point mutation	Point mutations	N/D *
17	Endocervical	Sensitive	Point mutation	Point mutations	N/D *
18	Endocervical	Sensitive	Point mutations	Point mutations	Point mutation
19	Urethral	Inconclusive	Point mutation	Point mutations	Point mutation
20	Urethral	Point mutations	Sensitive	Point mutations	Point mutation
21	Urethral	Inconclusive	Sensitive	Sensitive	Sensitive
22	Urethral	Point mutations	Sensitive	Point mutations	Point mutation

* N/D = No data, no band on agarose gel.

**Table 3 ijms-24-07218-t003:** Clinical validation results (95% confidence interval) after the study of the discordant samples.

	TP (N)	TN (N)	FP (N)	FN (N)	Sensitivity % (95%CI)	Specificity % (95%CI)	PPV % (95%CI)	NPV % (95%CI)
SpeeDx	36	56	3	7	83 (69–93)	95 (86–99)	92 (78–97)	88 (77–95)
Allplex	42	63	0	2	95 (84–99)	100 (94–100)	100 (89–100)	96 (88–99)
Viasure	43	63	0	1	97 (88–99)	100 (94–100)	100 (89–100)	98 (90–99)

TP: True positive; TN: True negative; FP: False positive; FN: False negative; PPV: Positive predictive values; NPV: Negative predictive values.

**Table 4 ijms-24-07218-t004:** Main characteristics of the three assays under study, ResistancePlus^®^ MG panel kit (SpeeDx assay), Allplex^TM^ MG & AziR Assay (Allplex assay), and VIASURE macrolide resistance-associated mutations (23SrRNA) Real time PCR detection kit (Viasure assay).

	SpeeDx Assay	Allplex Assay	Viasure Assay
Targets	MG + 5 rRNA macrolide resistance mutations	MG + 6 rRNA macrolide resistance mutations	WT region + 5 rRNA macrolide resistance mutations
Reagent format	Three separated tubes at −20 °C	Three separated tubes at −20 °C	Stabilized and ready-to-use strips
Instrument	Any instrument supporting Taqman technology (^a^ Cobas Z 480 PCR analyzer)	CFX96™ real-time PCR system	Any instrument supporting qPCR technology (^a^ CFX96™ real-time PCR system)
Pre-assay supplies	Color compensation	Not needed	Not needed
^b^ Turnaround time	1 h 15 min	2 h	1 h 30 min
Interpretation software	UgenTec FastFinder	Seegene Viewer	Open format (RealTime_PCR v7.9.)

MG: *Mycoplasma genitalium*; WT: Wild type; ^a^ instrument used in this study; ^b^ excluding DNA extraction.

## Data Availability

The data that support the findings of this study are available from the corresponding author (M.P.P.) upon reasonable request.

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
