# Peer review of "Retrospective and Comparative Study of Three Molecular Assays for the Macrolide Resistance Detection in Mycoplasma genitalium Positive Urogenital Specimens"

_ijms, 2023, doi:10.3390/ijms24087218_

Round 1

Reviewer 1 Report

I have sent it to the editor.

The study appears to be well-conducted; however, I have some concerns regarding the limitations of the study. Firstly, as pointed out by the authors, the study only involved a single centre, which limits the generalisability of the findings. Secondly, relevant patient information such as demographics, clinical history, and treatment regimen was not obtained, which could have confounded the results.

Moreover, the authors provided no novel insights or significant contributions to the existing literature. As they pointed out, several multi-centred prospective studies have already compared commercial detection kits for macrolide resistance in Mycoplasma genitalium and have obtained more robust and reliable findings.

Overall, while the study is well-conducted, the limitations and lack of novelty in the findings make it difficult to recommend for publication. I would suggest addressing the limitations and conducting a more comprehensive and novel study before submitting the manuscript for publication.

Reviewer 2 Report

This manuscript aimed to assess three molecular assays for the macrolide resistance detection in Mycoplasma genitalium positive urogenital specimens.

This manuscript is very well written and address very important point in the field. The results and discussion sections are also very well written based on the collected data.

However, some detail about detection of the bacterial should be mentioned in the methods section line 186.

Was there any variability in the bacterial concentration (ct value) based on the initial microbial identification? How this correlate with the results of the resistance?

It is not clear what the traditional methods of resistance detection that the authors used to compare those three methods to.

What is the cost of those methods compared to the traditional methods and how easily these methods can be implanted on large scale.

I think acknowledging these comments, I believes the manuscript would be suitable for publications after minor revision.

Round 2

Reviewer 1 Report

Introduction

Line 42: Mycoplasma genitalium is a Sexually Transmitted Infection (STI) recognized as cause of urethritis and further urogenital syndromes.

Change infection to organism.

Results

·       Number the tables in order.

Table-1 is in the Materials at the end of the article.

·       Add footnote indicators in Tables 2 & 4

·       Line 144: detection point mutations at positions 2058 and 2059 have not been mentioned in the results.

Recommended to add the position of the mutations detected by each assay in the results.

·       Line 151: (mutation no detected by SpeeDx and Viasure assays)

                Change to not detected

Materials and Methods

Line 210: the authors mentioned that the bacterium was detected during routine Laboratory diagnosis using the Allplex kit. Did they detect any mutation as well in the routine Lab?

It would be good to know and compare that to the data obtained in this study.
